# The Cult of the Child: A Critical Examination of Its Consequences on Parents, Teachers and Children

**Serge Dupont** [1,2,*] **, Moïra Mikolajczak** [1] **and Isabelle Roskam** [1]

1    Department of Psychology, 10 Place Cardinal Mercier, Université Catholique de Louvain,
     1348 Louvain-la-Neuve, Belgium; moira.mikolajczak@uclouvain.be (M.M.);
     isabelle.roskam@uclouvain.be (I.R.)
2    Deparment of Pedagogy, Haute École Galilée, Rue Royale 336, 1030 Brussels, Belgium
*    Correspondence: s.dupont@uclouvain.be

**Abstract:** The concept of the "cult of the child" highlights a radical change in child representation. Having been neglected and even disrespected for centuries, children are now valued, and their interests are placed above all others. This change in views of children, reflected in changes in laws, institutions and practices, has also spread to two pillars of our democratic societies, the family and the school, with a number of consequences for parents, teachers and children. The purpose of this article is to (1) describe the changes in thinking that have led to the cult of the child, (2) examine their consequences for children and parents, (3) examine their consequences for students and teachers, and (4) reflect on how to preserve the benefits of these changes while limiting the negative consequences.

**Keywords:** burnout; authoritative; permissiveness; needs; protection

## 1. Introduction

Intergenerational studies have presented a worrying picture of the new generation (i.e., iGen/Gen Z, born 1995–2012). Since 2010, researchers have observed a decline in mental health indicators in the United States, i.e., decreased happiness and increased loneliness, anxiety and depressive symptoms (Duffy et al. 2019; Twenge et al. 2018), a decline in vocabulary skills (see (Andreu and Steinmetz 2016) for France and (Twenge et al. 2019) for the United States) and even mixed attitudes similar to those of previous generations toward environmental protection in both Europe and the United States (e.g., Gray et al. 2019; VanHeuvelen and Summers 2019).

One explanation for this trend can be found in the intensive use of new technologies by IGen, born precisely at the time of the rise of social media. Many studies have shown that excessive use of smartphones and social networks has a negative influence on these mental health and cognitive indicators (for a review, see Twenge 2020). This effect is direct, but also indirect, via a disruption of in-person social interactions, interference with sleep time and quality, exposure to a toxic online environment and reduction in time spent in reading books (Twenge 2020; Twenge et al. 2019). However, other recent cultural changes, that may have been neglected so far, can also explain these generational changes, particularly in the educational sphere.

In the current article, we propose that this worrying situation of the new generation could be in part the consequence of what we call, inspired by historian Boas (1966), the "cult of the child", a recent phenomenon that places the interests of children above all others. We suggest that this cult leads to three attitudes towards children: (1) a decrease in the constraints imposed on them; (2) a concern to meet their every need; and (3) an attempt to prevent them from any harm or danger. We do not mean that placing interest in child needs is negative per se; we instead think that this excessive concern, which is becoming more and more popular, may affect mental health, cognitive skills and even the physical health of young people today.

The cult of the child is the consequence of a shift in the representation of children whose roots lie mainly in the 18th century and that has been accelerated in the last two decades through a series of laws and agreements aimed at protecting children's interests. This has changed not only parental practices but also those of teachers in schools, with potential consequences for children and adults, and for society.

In this special issue article, we aim to understand this historical development, to show how it has changed the relations between children and adults at the beginning of the 21st century and to reflect on how to preserve the advantages of the change in the way we think about children while avoiding its current pitfalls. We begin by describing the shift in representations of childhood from a historical perspective. Then, based on research in developmental and educational psychology, we examine the possible consequences of this cult for children and parents and for students and teachers. In the discussion, we reflect on ways to avoid current pitfalls and take a more intercultural perspective and end by underlining the limits of our approach. This narrative review is based on a selection of studies relevant to our theme of interest. No specific criteria for inclusion and exclusion have been pre-defined (Collins and Fauser 2004). Our objective is to stimulate scientific debate and open up new research perspectives.

## 2. Historical Background

The Ancient Greeks mainly regarded children as physically weak, morally incompetent and mentally incapable (Golden 2015). Aristotle, for example, regarded them as brutes who only pursued their own pleasure (Boas 1966). As a consequence, the main objective of the Greek education was to force the child to become "other". The nature of this other depended on the ideal help up by each city-state: the obedient soldier in the case of Sparta or the enlightened citizen in the case of Athens. Educational institutions, which were created at this time, were all centred around these objectives (Marrou 1982). In other words, there was no concern for children's well-being at that time—something that is underlined by the prevalence of violence as an everyday reality in education (Marrou 1982).

Catholicism historically viewed children in the same way as the Greeks. The infant Jesus was considered an exceptional being, so his veneration did not extend to other children. According to Boas (1966), there was no idealization of the child until the 18th century. The philosopher Pascal likened children to animals and the humanist Erasmus defined them as "half-human" creatures who must be corrected through intense discipline so as to prevent their bestial nature from appearing (Kennedy 2006).

The real and profound change in mentality came with the publication of Rousseau's book *Emile, or on education* (Cunnigham 2020; Koops 2012). Rousseau ([1762] 2010) depicted adults as spoilt by the artifices of civilization and close to death, whereas children were presented as lively, animated and enjoying the fullness of life. According to Rousseau, "Everything is good as it comes from the hands of the Maker of the world, but degenerates once it gets into the hands of man (p. 7)", and education should thus preserve this set of qualities rather than force the child to become other. It was no longer the mature and cultured man who was the model, but the innocent, pure, creative and curious child. To this end, Rousseau ([1762] 2010) proposed major pedagogical innovations based on freedom and the need to start from the child's interests and needs. Traditional education based on harsh discipline was rejected by Rousseau. Instead, he encouraged mothers to care for and be tender with their children (Koops 2012).

The Romantics in the 19th century consolidated Rousseau's vision of childhood (Boas 1966), as can be seen in many poems and tales of that period. This shift in the representation of children can also be seen from the paintings of this period (Koops 2003).

More recently, experts in education have recommended a new, gentler and more respectful way of raising children. Previously, parents were told to put their children to bed and feed them at regular, imposed times and to raise them "the hard way" to face the world. The works of the paediatrician Spock in the United States (Spock 1946) and of neo-Freudians such as Dolto in France (Dolto 1985) and Brown in the US (see Boas 1966)

take the opposite view. They invite parents to consider their children as individuals to whom they should listen and with specific needs that should be satisfied. Since Spitz's work on hospitalism (Spitz 1947), experts have increasingly insisted on the importance of the child's emotional needs and the need to decode and understand these needs.

In the post-Romantic period, pedagogues such as Pestalozzi, Fröbel or Key put forward severe criticisms of traditional teaching (Boas 1966), which they described as annihilating the child's potential: "the ability to act by oneself, the gift of observation, all qualities that children bring with them to school, have, as a rule, disappeared at the end of the school period" (Key 1909, p. 88). If children have lost their intrinsic qualities by the time they have finished school, this is because they have been forced to eat the poisoned fruit of the tree of knowledge (Boas 1966). Dewey ([1916] 2007) shared the same view, describing the traditional teacher as a "dictator" who mutilated the child's abilities. This critique was accompanied by concrete proposals: from now on, school must take into consideration the needs of each child by proposing activities—such as free play or experimentation—that would allow them to bloom.

The image of the child has thus been transformed. These young creatures with great intrinsic qualities must now be protected, listened to and cherished. This new status took concrete form during the 20th century through a series of laws and agreements aimed at protecting children's interests. In the next two sections, on parenting and school, we examine how the cult of the child emerged, how it became enshrined in law, and what its consequences are for practices, for children and for the adults who educate them. We have chosen to examine parenting and school because they are two pillars that shape the adults of tomorrow.

### 3. The Cult of the Child in Parenting

*3.1. The Advent of Child-Centered Parenting*

Until the 19th century, the family environment was considered as a place of intimacy with the paterfamilias (who held power over his wife and children) at its head. The state did not intervene.

At the end of the 19th century, children become of greater interest to the state. Poverty, crime and vice were widespread in industrialized cities, and the state began to see children as the future of the nation (Bullard 2015). It became necessary to protect them, not for their own sake, but to change and protect society as a whole. The idea began to gain ground that parents were responsible for the future of their children and that the state could intervene (by force) in families who were unable to ensure that their children survived and were brought up properly (Michel and Varsa 2014).

At the beginning of the 20th century, the First World War led to a fall in the birth rate, leading to a concerted drive to reduce infant mortality. Mothers were perceived as key to children's survival and were therefore encouraged to adopt new hygiene practices and to breastfeed their infants. Because children not only needed to survive but also represented the future of the nation, concrete measures were taken (King 2016), such as regulation of child labour (Somavia 2002), compulsory schooling (Miller 1989), child protection laws (Walsh 2020), the creation of juvenile courts (Thompson and Morris 2016), or the implementation of youth assistance or family support services through which the state could intervene in families that were deemed defective in order to protect children (Walker 2012). At the international level also, crucial political changes occurred, with the recognition for the first time of specific rights of children and the responsibility of adults in upholding these (League of Nations 1924), the creation in 1946 of the United Nations International Children's Fund (UNICEF), the Universal Declaration of Human Rights in 1948 (United Nations 1948), the Declaration of the Rights of the Child (United Nations 1959), and ultimately the International Convention on the Rights of the Child (CRC) (Convention on the Rights of the Child 1989). Besides instituting the rights of children, the CRC outlines the duties of those responsible for them, particularly parents, and the importance of the

role of signatory states in supporting parents in their complex task, monitoring them, and intervening (by force if necessary) when parents are not acting in the child's "best interest".

At the beginning of the 21st century, the Committee for European Social Cohesion published a report aimed at identifying the implications of the CRC for parenting (Daly 2007). This report sought to define the notion of "good parenting" using the concept of "positive parenting". The experts who participated in this committee relied on the scientific knowledge at that time about child development to provide guidelines on the parenting practices to be favoured or proscribed (Daly 2007). They were mainly influenced by two particular fields of research. The first was attachment, where research emphasized the importance of emotional security for optimal social, cognitive, affective and physical development (e.g., van Ijzendoorn et al. 1995), the influence of the caregivers' sensitivity and mentalization competencies, and the lifelong consequences of attachment failure for the child's development and health (e.g., De Wolff and van Ijzendoorn 1997). The second field was that of parenting styles. In reference to Baumrind's seminal work on parenting (Baumrind 1971), the authoritative style (as opposed to authoritarian, indulgent and neglectful styles) was associated with better cognitive, academic, emotional and social development, as well as with greater well-being and health for children. Accordingly, the combination of warmth and democracy seemed to be the formula for optimal parenting.

In this context, laws aimed at regulating parenting started to be introduced. The most emblematic example is undoubtedly the prohibition of spanking, adopted in 37 countries in the last 40 years (Global Partnership to End Violence against Children 2021). Never before in history has parenting been so socially regulated. Parenting is no longer a matter of common sense: it requires reference to the rules enacted by the state and to the knowledge disseminated by experts. These experts hold the keys to positive parenting that respects the rights of children and their best interests and seek to optimize their development. All the conventions, reports, rules and laws described in this section testify to the cult of the child and are the direct product of the development in the representation of the child outlined in the historical background section.

### 3.2. Development of Parenting Practices

The institutionalization of childhood in international conventions and national laws has led to changing parental practices. This change has occurred in just two generations and has been driven by experts, including both scholars and practitioners, through (1) the production of scientific knowledge about parenting and its relations with child developmental outcomes (Bornstein 2019); (2) the transmission of this knowledge in the form of formulas, recommendations or even injunctions to parents to adopt optimal childrearing behaviours (transmitted through online information, websites, popular scientific books, campaigns, etc.) (Kay 2010); (3) the assessments of parenting practices and the quality of the parent-child relationship by experts working in specialized consultation centres, when the child's development is not considered optimal (Foran et al. 2020); and (4) the growing number of preventive or curative parenting programs, under the guidance of experts or even self-administered, aimed at training parents to adopt optimal practices with respect to their children (e.g., Webster-Stratton 2005). Clearly, parenting experts have become important figures in the educational landscape today (Lee et al. 2014).

It is difficult to objectively state the extent to which parenting practices have really evolved over the last few decades. However, in order to illustrate the cult of the child, i.e., how the changes that have occurred have led to both a decrease in constraints and an increasing concern to meet all children's needs and avoid any dangers, we have gathered a set of indicators from several sources.

### 3.2.1. Changes in Parenting Practices towards Decreased Constraints

Classifications of parents according to the most documented parenting styles (i.e., authoritative, authoritarian) show an increase in the combination of warmth and democracy, with more parents being classified as authoritative, and a decrease in parents' authority,

with fewer parents being classified as authoritarian. We searched the PsycInfo and PsycArticles databases with the keywords Baumrind*authoritarian*authoritative*cluster. We selected studies published in peer-reviewed journals after the seminal work of Baumrind (1971) and Maccoby and Martin (1983), in which a classification of parenting styles from Western samples was reported. This bibliographical research is not intended to be exhaustive and is not without limitations. It does, however, give an initial hint as to the development of parenting practices. As the frequencies in Table 1 suggest, the percentage of parents classified as authoritative is tending to increase. It averaged 58% (min. 53%–max. 68%) in the studies up to 2000 and 75% (min. 57%–max. 88%) in the studies published in the 21st century, an average increase of 23%. In addition, the percentage of parents classified as authoritarian seems to be decreasing over time. It averaged 42% (min. 32%–max. 47%) in studies up to 2000 and 33% (min. 12%–max. 45%) in studies published in the 21st century, an average decrease of 9%.

**Table 1.** Frequencies (%) of parents classified under the authoritative and authoritarian parenting styles from 1987 to 2020 [1].

|  | **Authoritative** | **Authoritarian** |
|---|---|---|
| (Dornbusch et al. 1987) [2] | 53 | 47 |
| (Lamborn et al. 1991) | 68 | 32 |
| (Power et al. 1992) [3] | 53 | 47 |
| (Shucksmith et al. 1995) | 61 | 39 |
| (Aunola et al. 2000) | 55 | 45 |
| (Metsäpelto and Pulkkinen 2003) | 69 | 31 |
| (Wolfradt et al. 2003) | 67 | 33 |
| (Lee et al. 2006) | 57 | 43 |
| (De Bourdeaudhuij et al. 2009) [4] | 72/74/83/79 | 28/26/17/21 |
| (Garcia-Espana et al. 2009) | 86 | 14 |
| (Howenstein et al. 2015) | 88 | 12 |
| (Kuppens and Ceulemans 2019) | 81 | 19 |
| (Parra et al. 2019) [5] | 68/69 | 32/31 |
| (Calders et al. 2020) | 80 | 20 |

[1] Frequencies did not always add up to 100% because some authors use additional clusters that differ from one study to another (e.g., unclassified). For the purpose of comparison between studies, we have scaled the frequencies to 100%. [2] The authors report the classification of 50% of the sample as pure cases, but for comparison purposes, the frequencies have been recalculated to 100%. [3] The study was conducted in Japan and the United States; the frequencies reported are those for the United States. [4] The study was conducted in Spain, Portugal, the Netherlands and Belgium; four values are reported. [5] The study was conducted in Spain and Portugal; two values are reported. More than 50% of the data were missing, but for comparison purposes, the frequencies were recalculated to 100%.

Although the authoritative style is widely accepted by most scholars and experts as optimal (Daly 2007; OECD 2020), a certain shift towards an approach known as "exclusively positive parenting" (EPP) has been noted among parents and popularized in lay books in very recent years (e.g., Ducharme and Beaumont 2017). EPP is a parenting style almost exclusively based on warmth, support and autonomy of the child, avoiding any form of punishment (such as timeouts or intentional ignoring), discipline or even structure (considered as restrictive to the autonomy and exploration of the child). Although it contradicts the scientific evidence—i.e., authoritative style including firm discipline when needed (Larzelere et al. 2017), EPP is presented as the only possible alternative to maltreatment or downright violence towards children.

Though not without methodological limitations, studies of the intergenerational transmission of child-rearing practices also tend to confirm the development of parenting practices towards fewer disciplinary constraints. Although the correlations between successive generations are rather modest—suggesting that some transmission occurs but that there is also some discontinuity (e.g., Bailey et al. 2009)—intergenerational studies suggest greater continuity for practices such as warmth that have consistently been regarded as

desirable, than for those that have been regarded as more controversial in society, such as strict discipline (Roskam and Stievenart 2013).

### 3.2.2. Changes in Parenting Practices towards Meeting All Children's Needs

In order to meet their children's needs, parents are expected to be more involved in their parenting role, and in particular to spend more time with their children and be fully emotionally available to them. As early as the end of the 1990s, mothers who were interviewed about their parenting role already reported feeling increasingly under pressure to meet the cultural norms of highly involved parenting aimed at optimizing the child's development (Hays 1996). Hays (1996) proposed the notion of intensive parenting to designate the child-centred, expert-guided, emotionally absorbing, labour-intensive, and financially expensive approach to parenting that began to appear in the 1990s. Intensive parenting has been illustrated by many qualitative analyses of parents' experiences (e.g., Gomez Espino 2013) and confirmed by the increasing amount of time spent daily with children by their parents. In 11 Western countries, between 1965 and 2012, this time increased by 48% for mothers, from 54 min to 104 min on average per day, and by 73% for fathers, from 16 to 59 min (Sani and Treas 2016). This time spent with children has not only quantitatively augmented but is also increasingly used to optimize child development rather than for passive supervision (Craig 2006). While the intensification of parenting has often been considered more typical of mothers, fathers are increasingly involved too (Shirani et al. 2011).

### 3.2.3. Changes in Parenting Practices towards Avoiding Any Danger

In the 21st century, parents are well informed by experts of the dangers that threaten children. Therefore, to ensure their normal or even optimal development, parents are expected to avoid exposing their children to any known threat, e.g., smoking or drinking alcohol during pregnancy; not breastfeeding; providing food rich in salt, sugar or fat; using strict discipline, exposing children to technology (screens, 5G, cell phones, social media, etc.); or allowing them contact with people who may be a bad influence (Faircloth 2014). Parents who, despite knowing about these dangers, do not take all the necessary precautions to avoid them, are held morally responsible for any consequences. This includes not only about protecting children from immediate dangers present in the environment, but also predicting and preventing any circumstances that might interfere with the child's optimal development (Wolf 2011).

This risk aversion may lead to daily concerns, loss of self-confidence and increased stress in parents. These beliefs have been shown in patterns regarding changes in behaviours. For example, can they still let their children go to school on their own knowing that there is a risk that they will be kidnapped? Studies report that in 1969, 48% of American children walked or cycled to elementary school, whereas in 2009, only 13% did so (Twenge 2017). Moreover, 76% of parents nowadays report that they always know where and with whom their young adolescent is (against 62% in 1999; Twenge 2017). And geographical studies attest to the withdrawal of children from the streets of Western cities, with parents nowadays preferring to watch over them in person (Holt 2011).

Although risks that could potentially interfere with children's optimal development are indeed present in the environment, there is some disparity between the actual occurrence of certain phenomena and the concerns expressed about them (concerning breastfeeding, for example: see Wolf 2007). Thus, certain parental behaviours considered in the 20th century as normal practices or even good practices encouraging autonomy in children (e.g., letting them go to school on their own) are seen in the 21st century as benign neglect, i.e., neglect due to ignorance of the risk. As a corollary, parents considered in the 20th century as "good enough parents" are seen in the 21st century as "at-risk parents" who need to be advised or even trained by experts, and subject to state monitoring. The professionalization of parenthood is thus legitimized and "the task of raising children is turned into a skill" (Macvarish 2014, p. 99).

### 3.3. Consequences of the Cult of the Child for Children

The changes in institutions and practices described above were all made with a view to the best interest of the child. They have in fact had a number of positive effects, including the prohibition (moral or even legal) of all forms of violence against children, the reduction in the risk of unintentional injuries and falls in childhood, and the related mortality (Grossman 2000), more inclusive education with a greater tolerance for "non-normative" behaviour (e.g., homosexuality), and the increase of intimacy within families and in particular of parent-child quality time (Collishaw et al. 2012). However, and unfortunately, there have also been a number of negative consequences, which are becoming even clearer and more problematic as the cult of the child intensifies. In this section, we focus more specifically on the negative consequences of the above-mentioned changes, for both children and parents.

Although the beneficial effects of positive parenting on children have been widely documented (for a review, see OECD 2020), the intensification of positive parenting may have become counterproductive. While positive parenting aims at optimizing the development and well-being of the child, its intensification and excessive approaches to which it can sometimes lead, such as EPP or hyper- or over-parenting, have negative consequences for the child (Faircloth 2014). The system thus ends up working against those it claims to protect. The negative consequences have been empirically demonstrated in some recent studies that we selectively review below.

We will base our review on the consequences of "helicopter parenting", a typical form of parenting embedded in the cult of the child as characterized by the alleviation of disciplinary constraints and frustrations, the prioritization of meeting all the child's needs, and the focus on child protection. Helicopter parenting is by far the most studied of all the parenting styles that have emerged from the cult of the child. The term was coined in 1969 (Ginott 1969) and has become increasingly popular in the Unites States since 1990 (Cline and Fay 1990). It describes parents who are excessively child-oriented, over-involved, over-caring and over-protective. These parents disagree with the idea of their child being exposed to any risk and therefore behave in an intrusive and controlling way. Prevented from facing any problematic situation and therefore from finding solutions by themselves, overparented children have been called the 'cotton wool kids' (Bristow 2014).

Interestingly, and somewhat paradoxically, helicopter parenting has negative effects on the child's physical health: over-protective parents severely limit the child's possibilities of exploration, resulting in a reduction in motor activity (Janssen 2015). Helicopter parenting also has deleterious effects on the psychological well-being and mental health of children (Kouros et al. 2017). Children of helicopter parents are more anxious (Spokas and Heimberg 2008) and use more medication for depression and anxiety (LeMoyne and Buchanan 2011). They also report higher worries and psychological difficulties in emerging adulthood (Segrin et al. 2015).

Given its excessive focus on the child, it is not surprising that research has found that helicopter parenting is associated with narcissistic traits in children (Eberly-Lewis et al. 2018) and ego inflation (Yılmaz 2020). Moreover, by trying to anticipate and solve difficulties before they affect the child, helicopter parents prevent their children from becoming independent and making autonomous choices (Schiffrin et al. 2015). As a result, these over-parented children consider that they have the right to expect others to solve their difficulties and give them a lot of support of the kind that they received from their parents, creating a general sense of entitlement (Segrin et al. 2012). These children display a more external locus of control (Spokas and Heimberg 2008), procrastinate more (Hong et al. 2015), and show a lower level of school engagement (Padilla-Walker and Nelson 2012).

The finding that over-parented children show lower school engagement (Padilla-Walker and Nelson 2012) is somewhat paradoxical, because another facet of over-parenting is overstimulation. In order to optimize their child's development, parents provide numerous structured activities aimed at developing language quality, cognitive reasoning and so on. To achieve their goal, parents use stimulating materials at home, interact actively with the child's school experience and limit play and informal leisure time in favour of learning

time. While stimulation through participation in structured activities is important for child development, over-stimulation (i.e., excessive stimulation through too many structured activities) may have negative consequences by depleting children's energy, by not teaching children to organize their time by themselves and by exposing children to extremely demanding standards. Although the links with parental over-stimulation have not been formally demonstrated, it is likely that changes in parenting practices have contributed to the increase in perfectionism among young people in recent years. A meta-analysis gathering data from more than 40,000 college students from the United States, Canada and the United Kingdom between 1989 and 2016 has shown that young people perceive others as more demanding of them and are more demanding of themselves and of others (Curran and Hill 2019).

*3.4. Consequences of the Cult of the Child for Parents*

The increase in parental investment and pressure that has resulted from the cult of the child has most likely resulted in increased parenting stress and parental burnout. We describe this development only as a likelihood because, to the best of our knowledge, there is no cohort study on parental stress/burnout in lay parents. However, two indirect indicators suggest that parental stress and burnout may have increased over the past few decades. First, the notion of parental burnout, coined in 1983, has become increasingly popular with both the lay public and the scientific community, with a twentyfold increase in the number of publications using the term since the 2000s. Parental burnout (see Mikolajczak et al. 2021 for review) has recently been pointed to as an important research direction in psychology (Gruber et al. 2021). Interestingly, the prevalence of parental burnout has been found to be much higher in Western countries (Roskam et al. 2021), i.e., precisely where the cult of the child is the most apparent. The second indirect indicator that parental stress and burnout may be on the increase is a recent retrospective study by Mathy (2019) on 470 parents, suggesting that the prevalence of parental burnout in Belgium may have been eight times lower in the 1960s than nowadays. As the study was retrospective and conducted on a small sample, the results must be taken with great caution. However, they dovetail with those of the studies conducted in the school domain, where the cult of the child seems to have paved the way for teachers' burnout (see below).

*3.5. Summary*

As the foregoing shows, changes in the representation of the child have been paralleled with changes in parenting practices towards imposing fewer constraints on children and focusing more on their needs and protection. In parallel with this development, cohort studies found a host of negative outcomes for children, and parental burnout is now a hot topic. Based on this evidence, we can only speculate that the cult of the child has brought about these negative consequences. No study so far has directly tested the hypothesis of a direct link between the cult of the child and the negative outcomes observed in children and parents. The next section, showing the negative consequences of child-centred curriculums in schools, provides further evidence of the potential effects of the cult of the child and suggests that studies that directly and thoroughly examine its impact on parenting and schools are needed.

## 4. Cult of the Child in Schools

*4.1. Toward a Child-Centered School*

In the 19th century, school systems were established in Europe and the United States. Between 1852 and 1918, American states adopted diverse laws requiring school enrolment for children of 8 years of age and older (Mendez et al. 2017), and between 1881 and 1888, the French parliament approved a series of laws in favour of compulsory education, free primary education and secularism (Hirsch 2016). These projects were the direct result of the Enlightenment; they were aimed at creating citizens who were aware of public issues

and therefore able to take part in public government and debate, in accordance with the fundamental principles of emerging democracies.

At the same time, the child began to take its place in official texts and other pedagogical treaties. For example, the *Dictionnaire de Pédagogie* (Buisson 1911), written by a team led by Buisson, winner of the Nobel Peace Prize, founder of the League of Human Rights and director of French primary education between 1879 and 1896, served as a theoretical and practical guide for teachers (Nora 1997), and was partly based on contributions from Fröbel, Pestalozzi and Rousseau. Several articles reflected the desire to understand the nature of children in order to instruct them better. For example, the article on "school discipline" stated that "complete immobility and absolute silence are incompatible with the nature of childhood" (translated from the French, p. 200).

But it was only at the end of the last century that the child became the centre of some Western school systems. This child-centred school was promoted at the international level by the agencies mentioned above—the OECD, UNESCO and UNICEF—which worked for the acceptance of international conventions upholding the needs and rights of every child, such as the Dakar Framework for Action or the Millennium Development Goals. Their objectives were to advocate access to education for all and combat all forms of discrimination and violence against children, and also to promote child-centred teaching practices (Clair et al. 2012). In line with these principles, UNICEF issued the *Child-Friendly Schools Manual* (UNICEF (United Nations International Children's Emergency Fund) 2009), which states that (1) schools should operate in the best interests of the child; (2) educational environments must be safe, healthy and protective and (3) children's rights must be protected and children's voices heard. These points correspond to our conceptualization of the cult of the child. At the pedagogical level, UNICEF recommends that child-friendly schools should promote active and cooperative methods such as "discovery learning", that learning should be appropriate to the characteristics of each child and that students must be included in all aspect of school life (self-government).

These international agencies also defined the "good teacher". This teacher gives students more freedom, adjusts practices to the characteristics of each student, adopts active methods and is attentive to students' well-being. These criteria were used by the OECD in its Programme for International Student Assessment (PISA) and made it possible to measure the practices of teachers in order to verify how close they came to this ideal.

### 4.2. The Development of Teacher Practices

The classroom today is very different from that of the first public schools in France and the United States. The teacher who gives lectures and punishes the slightest deviation has been replaced by a more benevolent and caring figure, who leaves much more space and initiative to students. Like the changes in parental practices, these rapid changes can be explained in part by the role played by international agencies such as the OECD, UNESCO and UNICEF (Komatsu et al. 2021). Through PISA, the OECD offers each country a "diagnosis" of their education system (including an evaluation of teachers' practices) and then delivers a set of recommendations (for a review, see Pons 2017). These strategies of assessment, production and diffusion of scientific knowledge, advice and regulation have led, according to some researchers (e.g., Zapp 2021), to an international convergence in the pursuit of schooling goals and a gradual change in teacher training and practices, although this relation is not causal, and differences remain between countries (Pons 2017).

Although it is difficult to objectify these changes in practices—the studies measuring them are too recent—some data are available. For example, traditional practices such as constant monitoring and the extensive use of corporal punishment (Ariès 1973) are distant memories in most Western countries. In a study conducted by UNICEF and UNGEI (2019), less than one percent of 15-year-olds surveyed report having been violented by a teacher in Europe. In terms of pedagogical practices, there has also been, as recent PISA and TIMSS studies have shown, a decline in the West from transmissive practices to a student-centred learning approach that gives students more autonomy and more opportunities to discover

on their own or in groups, and takes their interests into account (OECD 2016), whereas teacher-centred practices remain dominant in Japan, for example (46% of the lesson time devoted to lecture-style presentation against 20% in England, Martin et al. 2008). The growing popularity of child-centred pedagogies such as "flexible classroom," which assigns each child a safe and individualized space in the classroom, is another clue to these changes in practices among teachers.

### 4.3. Consequences of the Cult of the Child for Students

In this section, we present a representative selection of psychological studies that have examined in greater detail the consequences of these student-centred practices.

### 4.3.1. Consequences of More Freedom for the Child

The "self-government" movement calls for children to be given more freedom and responsibility in schools. One area of research has examined the effects of giving students more or less freedom. Based on Baumrind's theoretical framework, this area focuses on classroom climate (authoritarian, permissive, autocratic or neglectful).

An authoritative school climate is characterized by a combination of strict but fair discipline and emotional support. This climate appears to reduce the proportion of school bullying (Cornell et al. 2015), violence toward teachers (Berg and Cornell 2016), depressive symptoms and substance abuse (Lau et al. 2017). In contrast, these factors increase in schools where autocratic (strict discipline without emotional support), permissive (emotional support without discipline), or neglectful (neither discipline nor emotional support) climates are observed. Interestingly, the highest scores for violence, bullying, depression and substance abuse are registered where the climate is permissive or neglectful (Lau et al. 2017). In short, these problems increase when discipline is absent.

### 4.3.2. Consequences of the Focus on the Child's Needs and Interests

The teacher today, encouraged by laws and international agencies, tries to respect the specific needs of each child. After all, numerous quantitative studies published in the field of motivational psychology have shown the positive influence of practices that respect the basic needs of autonomy, competence and belonging on both academic (i.e., school performance) and personal (i.e., self-esteem) factors (for a review, see Ryan and Deci 2020). The potential adverse effects of increased attention to children's needs have not yet, to our knowledge, been rigorously studied. However, some pedagogues in the United States have raised some concerns. Biesta (2017, cited in Floom and Janzen 2020) states: "A student-centred approach encourages that all of the attention be given to students' thoughts and feelings, with little mention of how these thoughts and feeling exist in relation to others and in the world around them. Further, when thinking about the world, child-centred learning creates a dynamic in which the children are relating everything back to themselves—an egotistical way of being". In line with this statement, recent preliminary findings collected in several countries (based on OECD data) have shown strong positive correlations between the student-centred approach and individualism (Komatsu et al. 2021).

This focus on child needs and interests has also led many Western countries to reduce the amount of knowledge in their curriculum and to replace it with activities that are more respectful of children's supposed nature (Hirsch 2016). However, cognitive psychology has shown that this strategy may hinder the development of skills that are considered crucial in the 21st century, such as reading comprehension and critical thinking (for a review, see Tricot and Sweller 2014). For example, readers who have knowledge of a topic consistently perform better in reading about it than those who are less informed; they also make more inferences and learn new words more easily (for a review, see Smith et al. 2021). To give a concrete example, to understand an article about the climate crisis, a person must be familiar with the concepts and methods specific to the discipline; this expertise will make it easier for this person to follow new events about this crisis and to distinguish between what is true and what is false (Byrnes and Dunbar 2014).

Based on this research, one may doubt the effectiveness of new schools' curricula that have emerged in this child-centred perspective. France provides a case study. The country was at the top of the international rankings in reading comprehension in 1991 (Raudenbush et al. 1996), just before the new school curricula, following the child-centred "Jospin Law", introduced in 1989, came into effect; in the latest international PIRLS study, France was ranked second-last in Europe (Colman and Cam 2017). National intergenerational studies have shown a similar decline (a 60% increase in the rate of errors on the same task) between 1987 and 2015 (Andreu and Steinmetz 2016). These studies further reveal a widening gap between students from advantaged and disadvantaged backgrounds, presumably because schools no longer compensate for initial family differences in knowledge (Gilkerson et al. 2017; Hart and Risley 1995). These curricula reforms are now the most compelling explanation for this phenomenon (Hirsch 2016).

### 4.3.3. Consequences of the Discovery Learning Approach

As stated above, the discovery learning approach has gained some acceptance in education policy in many Western countries (OECD 2016). However, the international PISA and TIMSS studies have shown that discovery learning is negatively associated with student performance (Hwang et al. 2018), whereas lecture-style presentation is positively associated with math and science performance (Schwerdt and Wuppermann 2011).

Alfieri et al. (2011) meta-analysed the results of 164 studies that compared the effectiveness of different instructional methods. Their results show that pure discovery (where students work on their own or in groups on problem-solving tasks, for example) is less effective for learning than enhanced discovery (with more guidance). These researchers also highlighted various factors that facilitate learning: precise feedback on the work done, working examples and scaffolding. Another meta-analysis by Stockard et al. (2018), which included 400 studies, confirmed the effectiveness of direct instruction on students' learning in different disciplines—reading, math or spelling—but also on the development of their self-esteem and on classroom behaviour. Finally, a study by Andersen and Andersen (2017), which included 56,000 Danish students, found that discovery learning increased achievement gaps between students from advantaged and disadvantaged backgrounds.

### 4.4. Consequences of the Cult of the Child for Teachers

The cult of the child and the associated pedagogical innovations place great demands on teachers: they must respect the needs and interests of each student, devise interesting activities and create a less structured classroom climate. The profession has become more complex and faces new demands. One of the possible consequences of this development is the increased incidence of burnout among teachers.

The first research on this topic dates back to the last two decades of the 20th century (for a review, see Chang 2009), precisely when children were acquiring a new status in many school systems; the number of studies then multiplied in the 2000s, possibly because of the high number of teachers affected by burnout. In a meta-analysis, García-Carmona et al. (2018) showed that 28.1% of the teachers' sample examined suffered from severe emotional fatigue, 37.9% from a high level of depersonalization, and 40.3% from a low level of personal accomplishment. This is especially worrying as teachers' burnout has been associated with absenteeism, difficulty in performing job duties, health problems (Chang 2009), and ultimately lower student performance (Madigan and Kim 2021). Burnout is also one of the reasons for which many teachers, especially younger ones, leave or consider leaving their profession (OECD 2018).

Researchers have identified many personal (e.g., personality) and organizational (e.g., class size) factors that lead to burnout. In a meta-analysis, Aloe et al. (2014) showed that disruptive students' behaviour was an important predictor of burnout among teachers. If we consider that in France, for example, 41% of secondary school teachers feel that their classes are disrupted by "a lot" of noise (OECD 2018); this gives an idea of how high the risk of teachers' burnout may be.

Based on this evidence, though, it is only possible to speculate that the cult of the child is responsible for this phenomenon. The hypothesis of a direct link between the two has not been directly tested so far.

*4.5. Summary*

Child-centred practices (self-government, a focus on students' needs and interests and discovery learning) are becoming increasingly popular in Western countries. The studies presented in this section show, however, that possible consequences of these practices are a decrease in student learning and performance and an increase in student depressive symptoms, disruptive behaviour and adherence to individualistic values. Teachers, on the other hand, may become exhausted in their efforts to get closer to students. Further research is needed to estimate more precisely the prevalence of the cult of the child in schools and confirm the associations presented in this section.

**5. Discussion**

Children are the product of our imagination (Koops 2003). If we perceive them as animals or sub-humans, we may legitimately be violent towards them, prevent them from expressing themselves or ignore their needs. Decades of research in psychology taught us that these strategies can seriously hinder children's development. Conversely, if children are seen as innocent, pure and curious, it is important to cherish them, to protect them and to satisfy their every need. In this article, we have argued that the latter view of the child, which has its roots in the 18th century and has had many benefits, has reached an extreme in the last two decades (an extreme that we have called the cult of the child) that may partly explain why the new generations show more symptoms of mental illness and less verbal competence than previous generations.

The fact that the cult of the child occurs both at home and at school is particularly noteworthy. First, it makes it difficult for parents and teachers to escape it, as each party will tend to reinforce it. Second, the fact that the cult of the child manifests itself in the two spheres in which the child spends the most time (i.e., family and school) probably increases its impact.

We also hypothesize that the cult of the child may affect the relationship between children and adults. Adults may become exhausted in their efforts to relate to and protect children, with potential consequences for their mental health, but also for the quality of the education they give to their children. In other words, the cult of the child may well backfire on adults too.

In addition to the above-mentioned consequences for children and adults, we identify an additional risk to be aware of. By getting too close to children, we risk neglecting the ultimate purpose of education: to create citizens. Nussbaum (2010) proposed that a citizen should be able to: understand the issues that affect the city and the world; criticize politicians who betray democracy; examine, argue and debate about different topics; recognize fellow citizens as equals; and ultimately prioritize the general interest over his or her own interest. It is only by producing citizens of this kind that a democracy will be able to face the main challenges of the present times. If the cult of the child creates individuals who are far from this ideal, i.e., more individualistic, more self-focused, less cultivated, and more reluctant to make an effort, this may complicate future struggles for a more sustainable, democratic and egalitarian society.

*5.1. Limitations and Directions for Future Research*

This article is intended as a first step in understanding the cult of the child and its possible consequences for children and adults. Further research is clearly needed to sustain our hypotheses. A first step would be to document the cult of the child more precisely. From a historical point of view, a comparative analysis of parental and teacher practices over time would inform us more clearly about this new view of childhood and its prevalence today. It is also necessary to examine the differences between socioeconomical levels and

between countries. The majority of the arguments presented in this article are justified via studies conducted among middle- and upper-class people from Western countries. It is likely that the cult of the child is much less present in countries with more collectivist values. We have seen, for example, that teaching in Japan is much more teacher-centred than student-centred (Martin et al. 2008). Similarly, parental burnout, which we assume is linked to the cult of the child, is higher in individualistic countries than in Japan (Roskam et al. 2021).

At the methodological level, we suggest creating measurement tools to identify and evaluate practices of parents and teachers that correspond to the three dimensions that we have identified in this article: (1) a decrease in constraints; (2) a concern to meet children every need; and (3) a concern to avoid any danger. These new tools would then allow for a more accurate examination of the associations between such practices and their consequences for children and adults.

We also recommend using longitudinal methods. For example, current studies tend to show that educational games (Ryan and Rigby 2019), a practice close to the cult of the child, because play is in children's nature, have positive effects on motivation and learning. However, what about the long-term effects? Is there not a risk that the child will have difficulty in the future taking an interest in something that does not match his or her immediate interests? It may also be because of the cult of the child that a decline in motivation during the school career has been observed in many countries (Scherrer and Preckel 2019). Ryan and Deci (2020) argue that this phenomenon is a consequence of the lack of consideration of students' basic needs. The data identified in this article suggest that the opposite hypothesis is plausible: if teachers make students the centre of attention too much, students may lose the will to engage with new subjects.

Finally, we recommend examining the interrelations between parental and teacher practices and how they may influence children's development. We have seen that helicopter parenting was associated with a decrease in student engagement. This may ultimately affect teachers' practices and mental health. Similarly, student-centred approaches may also affect the practices of parents, who may try to compensate for their children's deficiencies as students. As schools and parents are the two pillars that create future citizens, it is important to study these two domains in concert and from an interdisciplinary perspective.

*5.2. Implications for Parenting and Parent–Child Relationship in the 21st Century*

The "spirit of child protection" has certainly brought about salutary advances, but its extreme version—the cult of the child—may have many drawbacks. The ideal would be to find a fair balance between the immediate interests of the child and those of society. This new balance would require a change in parental practices that would take into account the data we have presented in this article. Concretely, and if the ultimate purpose is to create citizens in line with Nussbaum's conceptualization, we would encourage parents to:

- Take a long-term perspective. The immediate interest of the child is not always the interest of the future adult and of society.
- To balance children's needs with those of others and of the world around them. This entails restoring discipline and standing firm on certain key principles in line with a democratic, sustainable and inclusive society.
- Combine firmness with benevolence in order to achieve this balance. Far from being opposed to benevolence, firmness is its natural ally. Without it, benevolence is a source of insecurity for children, making them incapable of relating to others and their environment with respect (Larzelere et al. 2020).
- Let children breathe, have their own experiences and overcome difficulties without the stifling presence of parents.

The first three points also apply in a similar form to teachers. We further recommend that teachers focus more on knowledge, as it is essential for the development of 21st century skills such as reading comprehension and critical thinking. In addition, teachers should not underestimate their role. Explaining, modelling, giving feedback and providing

scaffolding are essential for student learning, especially for students from a disadvantaged background or with learning disabilities. Letting these students work on problem-solving tasks without guidance may well create a situation of cognitive overload that has deleterious consequences for learning (Kirschner and Hendrick 2020).

### 5.3. Contribution of This Article and Conclusion

The benefits of the shift in views of children over the last century appear so obvious that the possibility of negatives consequences of the current representations and practices (referred to collectively here as the "cult of the child") have been possibly underestimated and under-researched. The preliminary evidence gathered in this article suggests that research is urgently needed on this issue.

The cult of the child has been promoted at various levels of power in the hope of creating a more democratic and inclusive society. However, such an assumption is not borne out by the facts. Based on the evidence reviewed here, Emile, Rousseau's ward, is unlikely to become a citizen who is concerned with the issues affecting the city, who is critical and who puts the common good first. His most likely fate is to become immature, ignorant and selfish. If, on the other hand, we want parents and teachers to raise children to the rank of citizens capable of meeting ecological, economic, health and social challenges, the goal is to strike a fair balance between the interests of the child and the interests of society.

**Author Contributions:** S.D., I.R. and M.M. developed the general structure of the article and wrote collectively the general discussion. S.D. wrote sections "Historical Background" and "The Cult of the Child in school"; I.R. wrote the section "The cult of the Child in family" and coordinated the work; M.M. wrote the subsection "The consequence of the Cult of the Child on parent" and corrected several times the successive drafts. All authors have read and agreed to the published version of the manuscript.

**Funding:** This research received no external funding.

**Institutional Review Board Statement:** Not applicable.

**Informed Consent Statement:** Not applicable.

**Data Availability Statement:** Not applicable.

**Conflicts of Interest:** The authors declare no conflict of interest.

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
