# Peer review of "The Cult of the Child: A Critical Examination of Its Consequences on Parents, Teachers and Children"

_socsci, doi:10.3390/socsci11030141_

Round 1

Reviewer 1 Report

Strengths of the manuscript include the large literature revision of the topic “the Cult of the Child” throughout important developmental context such as family, peers, and school. The study is very interesting and could provide a relevant contribution to the understanding of this topic. The authors make an in-depth review of each of the contexts that undoubtedly help to clarify the meaning and consequences of the Cult of the Child.

---Specific comments---

  1. I appreciate the work by the authors but I wonder if the first step in order to broaden this topic it would do a systematic review or a metanalysis focus on the concept “the Cult of the Child” and not the work realized. I would be useful to find out if this work already exist and if not, that the authors consider doing it.
  2. In general, I noted that the literature review included too much cited outdated. I suggest review the studies cited and add those from five last years as APA style recommend.
  3. It is beyond my understanding why the authors use two different labels in the familiar context “family” and “parents” if they are later treated as synonyms throughout the manuscript.
  4. It is important that the authors review this question because the second objective of the study is “to examine its consequences on the family, children and parents”. Please review the manuscript or the second aim for flow and clarity.
  5. The Discussion section is quite brief and underdeveloped. I suggest that, once again, the authors review this section and make an effort to connect it with the information provided throughout the manuscript.
  6. At the end of the manuscript, the contributions of the study should be explained in details.
  7. Please, revise according to 7th Edition APA style all manuscript. Sometimes they cited in the wrong way, other times miss doi number. For instance, you have to add doi number in this reference: Demerouti, E., Bakker, A. B., Nachreiner, F., & Schaufeli, W. B. (2001). The job demands-resources model of burnout. Journal of Applied psychology, 86(3), 499.

Author Response

  1. I appreciate the work by the authors but I wonder if the first step in order to broaden this topic it would do a systematic review or a metanalysis focus on the concept “the Cult of the Child” and not the work realized. It would be useful to find out if this work already exists and if not, that the authors consider doing it.

Reply: To our knowledge, this is the first time that such a hypothesis has been put forward in these terms in the field of psychology. Whereas the term “Cult of Childhood” existed in the field of History, the term does not exactly refer to what we have labelled here “The Cult of the Child”. This is the first paper to introduce this term and to propose a definition of it. Thus, our work is better conceived as a first theoretical contribution, which will hopefully stimulate other research and allow a meta-analysis in a decade or so.

The originality of our work is precisely (1) to adopt an interdisciplinary approach to highlight this excessive focus on the child over the last two decades, (2) to introduce and define the concept of “Cult of the child” and (3) to examine its potential consequences for children and adults at the beginning of the 21st century.

The reviewer is right, however, that the specific contribution of our work was not sufficiently highlighted in the original version of the manuscript.

Changes in the manuscript: The introduction has been completely rewritten in order to clarify our approach, to underline the central hypothesis and to propose a conceptualisation of what we call "the Cult of the Child" which corresponds to a recent phenomenon that places the interest of the child above all others (see Introduction).

In the new version of the discussion, we also highlight the contribution and specificity of our work.

  1. In general, I noted that the literature review included too much cited outdated. I suggest review the studies cited and add those from five last years as APA style recommend.

Reply: In the original manuscript, we indeed relied for the historical part largely on classic works by historians, still cited today (Boas, 1966, Marrou, 1988). We also cited a number of older authors because their writings illustrate the spirit of the times (for example Dolto, 1986; Spock, 1946 or Key, 1909). In reviewing the literature of psychological studies, we were more careful to rely on more recent studies and systematic reviews. It is true, however, that more recent references were missing for each section.

Changes in the manuscript: We reviewed each out-dated reference to see if it could be removed and replaced with more recent work. This review led to the deletion of several references and the addition of new ones, such as:

For the historical part:

Cunnigham, H. (2020). Children and Childhood in Western Society since 1500 (3nd ed.). Routledge.

Golden M. (2015). Children and Childhood in Classical Athens. Johns Hopkins University Press.

For the institutionalization part:

Mendez, S. L., Yoo, M. S., & Rury, J. L. (2017). A brief history of public education in the United States. In R. A. Fox & N. K. Buchanan (Eds.), School choice: A Handbook for researchers, policy makers, practitioners, and journalists. Hoboken: WileyBlackwell.

Pons, X. (2017). Fifteen Years of Research on PISA Effects on Education Governance: A Critical Review. European Journal of Education, 52(2), 131–144. https://doi.org/10.1111/ejed.12213

Zapp, M. (2021). The Authority of Science and the Legitimacy of International Organisations: OECD, UNESCO and World Bank in Global Education Governance. Compare, A Journal of Comparative and International Education. 51(7), 1022-1041. https://doi.org/10.1080/03057925.2019.1702503

For the cult of the child in schools:

Kirschner, P. A., & Hendrick, C. (2020). How Learning Happens: Seminal Works in Educational Psychology and What They Mean in Practice. Routledge.

Madigan, D. J., & Kim, L. E. (2021). Does teacher burnout affect students? A systematic review of its association with academic achievement and student-reported outcomes. International Journal of Educational Research, 105, 101714. https://doi.org/10.1016/ j.ijer.2020.101714

  1. It is beyond my understanding why the authors use two different labels in the familiar context “family” and “parents” if they are later treated as synonyms throughout the manuscript.
  2. It is important that the authors review this question because the second objective of the study is “to examine its consequences on the family, children and parents”. Please review the manuscript or the second aim for flow and clarity.

Reply to comments #3 and #4: It is true that there was some confusion between the two terms, as noted by the reviewer.

Changes in the manuscript: we have now reviewed the manuscript and clarified the terms in the titles (Cult of the Child in parenting) and in the core text, referring to “Parenting” rather than to “Family/Families”.

  1. The Discussion section is quite brief and underdeveloped. I suggest that, once again, the authors review this section and make an effort to connect it with the information provided throughout the manuscript.
  2. At the end of the manuscript, the contributions of the study should be explained in details.

Reply to comments #5 and #6: we agree with the reviewer: the discussion was too short and did not sufficiently relate to the information developed in the manuscript. The main reason was that we were constrained by the word count limitation.

Changes in the manuscript: as we reduced some parts of the article (especially the part about the school), we were able to develop more largely certain aspects. More specifically: (1) we have expanded the sections: “Limitations and Future Directions” and “Recommendations for Practice” and (2) we have added the following section: “Contribution of this paper and Conclusion”.

  1. Please, revise according to 7th Edition APA style all manuscript. Sometimes they cited in the wrong way, other times miss doi number. For instance, you have to add doi number in this reference: Demerouti, E., Bakker, A. B., Nachreiner, F., & Schaufeli, W. B. (2001). The job demands-resources model of burnout. Journal of Applied psychology, 86(3), 499.

Reply: The reviewer is right: the references (in the text and in the bibliography) were not in the 7th Edition APA style.

Changes in the manuscript: All references have been corrected based on the latest APA recommendations.

Reviewer 2 Report

I found the MS very interesting and almost ready for publication.

However, in my opinion, some relevant caveats need to be considered by the Authors.

More specifically, I’m referring to the fact that nothing is reported about some crucial changes in the cult of the child due to the following socio-cultural events:

1) women’s emancipations and the new role into the family and parenting too;

2) Freud’s theory has significantly impacted the cult of the child starting from the beginning of the ‘900.

I suggest the Authors carefully refer to the previous work of Koops regarding the topic of this work.

See per example: Koops, W. (2012). 1. Imaging Childhood. In Beyond the century of the child (pp. 1-18). University of Pennsylvania Press.

Finally, please note some errors: the correct pronouns are both her and him when you refer to the child.

Author Response

Reviewer :

More specifically, I’m referring to the fact that nothing is reported about some crucial changes in the cult of the child due to the following socio-cultural events:

  • women’s emancipations and the new role into the family and parenting too;

Reply: There are indeed many factors or historical changes that impacted the evolution of parental practice. The one the reviewer points here had a complex role. Intuitively, one would expect that women’s emancipation rather had the opposite effect: a stronger focus on autonomy of kids, a certain neglect of their needs, more delegation to external agencies such as school and nursery. Indeed, a deeper understanding of what happened, why it was not the case, would be interesting, but it is beyond the scope of the present article.

  • Freud’s theory has significantly impacted the cult of the child starting from the beginning of the ‘900.

Reply: In his book “The Cult of Childhood”, the historian of ideas George Boas refers to the figure of Freud. But he points out that Freud had an ambiguous image of the child, which he often associated with the 'primitive', for example in his book Totem and Taboo. There is no sense of nostalgia for childhood in Freud's writings. Instead, Boas points the role of the neo-Freudians, who propagated the idea that education was 'a repressive force' that annihilates the child's potential. In the original manuscript, we mentioned the psychoanalysts Françoise Dolto and Brown who contributed to the success of this idea in France and US. This link between psychoanalysis and the evolution of the cult of childhood was, however, not sufficiently marked in the original manuscript.

Change in the manuscript: In the revised manuscript, we have given more emphasis to the influence of neo-Freudians on the education of children.

  • I suggest the Authors carefully refer to the previous work of Koops regarding the topic of this work.

Reply: We have read with great interest three articles by Willem Koops on the evolution of representations of the child throughout history. His work on the place of children in paintings over time confirms the idea of a change in the way the child is represented. This change became more pronounced after the writings of Jean-Jacques Rousseau. So we are particularly grateful for this suggestion: Koops allows us to give more strength to our historical argument.

Changes in the manuscript: In the historical background section, we refer to Koops' interpretation of Jean Jacques Rousseau's “Emile ou de l'éducation” and his work on the place of the child in painting over time. In the discussion, we mention, again based on Koops, that the child is the product of the adult's imagination and that this imaginary construction influences the way adults educate them and ultimately their development.

Reviewer 3 Report

Review for the MS “The Cult of the Child: A Critical Examination of its Consequences on Parents, Teachers and Children”

Date: 19.11.2021
Journal: Social Sciences
Publisher: MDPI
Round: 1

Dear Editor and Authors

I thank you for the opportunity. I feel the topic of the manuscript and the special issue are important and topical. So, I am very happy to contribute through my review. Let’s see if I can help the process. I will read this through and write my review and make a recommendation at the end.

****

After reading the manuscript, I can see a lot of potential in it. I agree the importance of this kind of narrative paper. It could provide an important synthesis about the change of the children’s role in the society through centuries. The approach via family, children and parents seems pragmatic and a solid decision. However, the reading raised many concerns that needs to be clarified before I can recommend anything else than decline for the manuscript. I will list my concerns below and let me hear what do all stakeholders think about them.

****

Concerns:

  1. The central concept “The Cult of the Child” has not been defined or addressed more than just giving the name. In addition, According to Boas (1966) it should be “the cult of childhood”.
  2. The English is very poor, and I am not a native English speaker. Is this manuscript a translation?
  3. The rationale is missing from the introduction, and there is a copy-paste stuff from the abstract. What is the motivation for the set objectives? How it is related to earlier research?
  4. I appreciate the government level guidelines and historical references, but I think it is very difficult to draw generalized perspectives on parenting in global scale through them. There are many cultures around the world that I recommend setting a lot of parameters for the analysis. For example, you know that schools are a place where young people meet other people right, and the hard work of learning is done during the free time. The family background is the number one factor that determines child’s future in e.g. academic perspective. Similarly, the substance use, alcoholism and poorness inherit from past generations. In some families, parents have tools cognitive tools to help kids how to learn and be persistence even with harder obstacles. Other families can support sports activities like wrestling that enable high-level pressure duration and kids can grow up with less anxiety of facing peer level violence etc. Other families can’t provide role models or offer hobby resources. Divorce number is over 50 % in e.g. Finland that brings one extra factor to the equation.
  5. The theoretical frame is heavy to read and complex to understand. One clear way how to improve the readability of theoretical sections is to implement infographics and tables that will produce a synthesis and show the central thoughts of the authors. I demand these skills and metalevel tools from all my students from the 1st year throughout their whole academic career.
  6. You mentioned data but this research doesn’t have any data. There is no data gathering or analysis described.
  7. The helicoptering parenting section is interesting. It is like a swearword in media and research. However, drawing out many different theoretical concepts and select studies to back up the arguments is one-sided way to build this narrative. I recommend authors to take a more critical approach in writing.
  8. The discovery learning part is a good example from this. If you could read some of the latest studies done in questions in science education field, you could see some evidence that the potential of young learners and discovery learning approach effects to creativity seems to be unlimited. With the current situation of wicked problems, the world needs thinkers with high-level cognitive skills, not emphasizing the lower levels skills.
  9. The future research recommendations are good. We need longer studies and comparative studies etc.

Overall recommendation: I recommend Decline for this manuscript.

Even though it has a lot of potential, I think the synthesis is too general and not justified via solid arguments. The phenomenon is more complex than this narrative literature-based approach can explain. I suggest more systematic approach. In addition, the motivation is not specified using earlier literature and the structure makes the manuscript difficult to understand. The whole manuscript needs to be build again before resubmitting.

Author Response

  1. The central concept “The Cult of the Child” has not been defined or addressed more than just giving the name. In addition, According to Boas (1966) it should be “the cult of childhood”.

Reply: We agree with the reviewer's remark that a weakness of our original manuscript was that we did not provide a univocal and clear definition of the concept of "the Cult of the Child". After all, this is probably the reason why there was some confusion with the Historian Boas’ Concept “Cult of Childhood” that we also cited. Whereas Boas “Cult of Childhood” has inspired our concept “Cult of the Child”, it does not exactly refer to the same idea. As you can read in the introduction, we now addressed this point more clearly, provided a definition of Cult of the Child”: the "Cult of the Child" corresponds to a recent phenomenon that places the interest of the child above all others. This cult would translate into three attitudes towards children: (1) a decrease in constraints imposed on them; (2) a concern to meet all their needs; and (3) an attempt to prevent them from any harm or danger - and specified our level of analysis, that is a psychological perspective with a focus on relationships between children and parents and between teachers and students. It is hopefully clearer now that we adopt a narrower focus of analysis than the one of Boas. The Boas “Cult of Childhood” presumes a more historical and macro level framework. Nonetheless, the Boas concept was a source of inspiration for us.

Changes in the manuscript:  The introduction has been completely rewritten and we have now defined the central concept of "the cult of the child" (see above).

  1. The rationale is missing from the introduction, and there is a copy-paste stuff from the abstract. What is the motivation for the set objectives? How it is related to earlier research?

Reply: We agree that we did not clearly link our objectives to earlier research.

Changes in the manuscript:  The introduction has been completely rewritten. Specifically: (1) we have modified the tagline by referring directly to the intergenerational studies carried out by Twenge and his colleagues (in order to relate to earlier works); (2) we have clarified our approach and objectives; (3) and we have defined "the cult of the child" (see Comment #1 above).

The resulting Introduction is now much clearer than in the previous version of our article.

  1. The English is very poor, and I am not a native English speaker. Is this manuscript a translation?

Reply: Our paper had been proofread by a native English speaker before submission but, in light of the Reviewer’s comment, we have decided to send the revised manuscript to a professional agency for proofreading.

Changes in the manuscript: the revised manuscript (not this letter, but the paper) has been proofread by the Translation/Proofreading agency Crossword.

  1. I appreciate the government level guidelines and historical references, but I think it is very difficult to draw generalized perspectives on parenting in global scale through them. There are many cultures around the world that I recommend setting a lot of parameters for the analysis. For example, you know that schools are a place where young people meet other people right, and the hard work of learning is done during the free time. The family background is the number one factor that determines child’s future in e.g. academic perspective. Similarly, the substance use, alcoholism and poorness inherit from past generations. In some families, parents have tools cognitive tools to help kids how to learn and be persistence even with harder obstacles. Other families can support sports activities like wrestling that enable high-level pressure duration and kids can grow up with less anxiety of facing peer level violence etc. Other families can’t provide role models or offer hobby resources. Divorce number is over 50 % in e.g. Finland that brings one extra factor to the equation.

Reply: The focus of the paper is the changes of the representation of child and its consequences throughout the late history of the western countries. It certainly does not have the ambition to generalize to other cultures or to be universal in its perspective. Your comment made us realize that we may have not been clear enough about this in the previous version of the paper. We acknowledge that it is important to be more explicit about this. Therefore, some sentences have been added in the paper to address this point more explicitly (see “Changes in the Manuscript” below).

Regarding the weight of the family: we have already pointed out that the family background has an influence on children's academic career, based on Hart and Risley (1995) and Gilkerson et al. (2017). However, when schools focus excessively on child’s needs and interests, there is a real risk of increasing these initial differences between children from different backgrounds. Teachers have also an important impact on student academic career, as it has been shown by John Hattie’s (2009) meta-analyses. In other words, family is one factor among other, obviously very important, but not the only one.

Changes in the manuscript. Throughout the article, we have been careful to make it clear that we are focusing on studies conducted in Western countries. Sometimes we refer to other countries (e.g. Japan) to show that there are important differences in parental or school practices.

In the discussion, we make it clear that our proposal is a hypothesis (a first step) that needs to be validated through comparative studies between countries and between families from different socioeconomical backgrounds. We also stress the need to study the interrelationships between family and school in order to better understand their influence on the child's academic career.

This new version seems to us to be much more nuanced than the previous version.

  1. The theoretical frame is heavy to read and complex to understand. One clear way how to improve the readability of theoretical sections is to implement infographics and tables that will produce a synthesis and show the central thoughts of the authors. I demand these skills and metalevel tools from all my students from the 1st year throughout their whole academic career.

Reply: we agree with the reviewer's observation that our theoretical development was quite complex.

Changes in the manuscript: we have lightened the theoretical frame (through several deletions of paragraphs) and also tried to better structure it. The thread has also been accentuated. These changes have increased the readability of the text and we hope that it is now more enjoyable to read. 

  1. You mentioned data but this research doesn’t have any data. There is no data gathering or analysis described.

Reply: The current paper is a narrative review. As such, and in order not to make the reading more cumbersome, we have not added too many data. However, we have been careful to remain faithful in our formulations to the effect sizes observed in the studies we mention.

Changes in the manuscript: some data have been added, in particular to give a more precise idea of the evolution of practices in the sections Cult of the Child in parenting and Cult of the Child in schools.

  1. The helicoptering parenting section is interesting. It is like a swearword in media and research. However, drawing out many different theoretical concepts and select studies to back up the arguments is one-sided way to build this narrative. I recommend authors to take a more critical approach in writing.
  2. The discovery learning part is a good example from this. If you could read some of the latest studies done in questions in science education field, you could see some evidence that the potential of young learners and discovery learning approach effects to creativity seems to be unlimited. With the current situation of wicked problems, the world needs thinkers with high-level cognitive skills, not emphasizing the lower levels skills.

Reply. We agree with this comment: the original manuscript may have given the impression of a lack of nuance. We also believe that the brief introduction and discussion contributed to this impression.

With regard to discovery learning: in the original article, we distinguished between two types of discovery learning: guided discovery and pure discovery, clearly showing that it was "pure discovery" (without any support from the teacher) that risked having negative effects on learning and widening inequalities between pupils from privileged and disadvantaged backgrounds.

Secondly, we fully agree with the reviewer's aim of education, namely to create citizens with high cognitive skills (as we pointed out in the discussion, drawing on the work of the philosopher Martha Nussbaum). However, as cognitive psychology shows, one should not confuse means and ends. It is not because a child is put in a research situation that he will become a researcher. It is not because we ask a child to be creative that he will become creative. One should not 'skip' steps, as this may be counterproductive (see for example: Kirschner, P. A., & Hendrick, C. (2020). How Learning Happens: Seminal Works in Educational Psychology and What They Mean in Practice. Routledge)

In the present paper, we are not arguing that children should be deprived of situations in which they are asked to be creative or critical (rather the contrary; we actually fully agree with the reviewer), but we are trying to make the point that the key is to find the right balance and not to underestimate the role of the teacher, especially when it comes to complex skills such as creativity or critical thinking.  

Changes in the manuscript: in the introduction we made it clear that the concept of "the Cult of the Child" was a hypothesis about a phenomenon that seemed to us to be gaining in importance in the last two decades. Then we specified our approach: we present a selection of research in educational and developmental psychology that corroborates this idea and that constitutes a first step in understanding the possible effects of this cult on children and adults. Our aim is certainly not to be exhaustive or categorical, but to open the scientific debate on this issue and to suggest new avenues of research.  

Secondly, throughout the article we have removed wording that we felt was too categorical and replaced it with more nuanced terms.

Finally, in the discussion, we emphasise that it is obviously necessary to conduct studies to validate our hypothesis.

Round 2

Reviewer 1 Report

Let me begin by thanking the authors for their effort revising this manuscript. The manuscript is much improved as a result of these edits. I believe that now the manuscript is ready to be accepted.

Author Response

We sincerely thank the reviewer for his previous comments that allowed us to improve our manuscript.

Reviewer 3 Report

Review for the MS “The Cult of the Child: A Critical Examination of its Consequences on Parents, Teachers and Children”

Date: 30.1.2022
Journal: Social Sciences
Publisher: MDPI
Round: 2

For Authors and Editor

I thank authors for their good work in preparing the revision. With the explanations the paper is easy to understand and can be accepted after minor changes.

The challenge is to explain this to readers who doesn’t have opportunity to see read the metatext behind the scenes. This aspect would be my recommendation for improvements in this second round.

*****

Concerns/comments for round 2:

  1. Could you explain to readers what your work’s relation to Boas (1966) term is “the cult of childhood”. You wrote that it has served as an inspiration.
  2. English is OK now.
  3. The rationale has improved.
  4. The theoretical framework ends to a summary 4.5. Could you elaborate the summary via a table or infographic including the references? It would demonstrate your ability to provide a synthesis.
  5. I still have an issue with the data mention because you do not gather, process or present any data.
    1. I recommend rephrasing “The majority of the data presented in this article comes from studies conducted among middle- and upper-class people from Western countries.” to “The majority of the arguments presented in this article are justified via studies conducted among middle- and upper-class people from Western countries.”
    2. In addition, I recommend stating clearly that this is a narrative literature review and explain to readers what kind of reliability and validity matters are needed to consider in using this approach. And a nice methodological reference would also be in order.

Overall recommendation: Minor revisions. Especially the research methodology needs to be opened to researchers.

Author Response

We would like to express our gratitude to the Reviewer for his/her comments and suggestions. Responses to reviewer comments :

  1. Could you explain to readers what your work’s relation to Boas (1966) term is “the cult of childhood”. You wrote that it has served as an inspiration.

Change in the manuscript : we clarify in the revised version that we draw on the work of Boas for our concept The Cult of the child : (34-35) (…) the consequence of what we call, inspired by classic work of historian Boas (1966), the "cult of the child", a recent phenomenon that places the interests of children above all others.

  1. The theoretical framework ends to a summary 4.5. Could you elaborate the summary via a table or infographic including the references? It would demonstrate your ability to provide a synthesis.

In the last version of our manuscript, we stated clearly that this is a narrative review and not a systematic one. Based on that, we do not think that there is any real added value in including a summary via a table or a graph. Summaries are proposed and have been approved by the other reviewers. We have made a real effort to be clear and precise in the previous revised version of the manuscript.

  1. I still have an issue with the data mention because you do not gather, process or present any data.
    1. I recommend rephrasing “The majority of the data presented in this article comes from studies conducted among middle- and upper-class people from Western countries.” to “The majority of the arguments presented in this article are justified via studies conducted among middle- and upper-class people from Western countries.”

We thank you for the suggestion. In line with the reviewer's comment, we have changed this sentence in the discussion.

    1. In addition, I recommend stating clearly that this is a narrative literature review and explain to readers what kind of reliability and validity matters are needed to consider in using this approach. And a nice methodological reference would also be in order.

In the case of a narrative review of the literature, there is no particular methodology, unlike a systematic review of the literature, which must be based on strict criteria for the selection and exclusion of studies, notably on the basis of their quality. A narrative review allows for a broader reflection, as is the case in our article, whereas a systematic review generally focuses on a specific topic ; as Collins and Fauser (2004) state : Narrative reviews generally are comprehensive and cover a wide range of issues within a given topic, but they do not necessarily state or follow rules about the search for evidence. Also, typical narrative reviews do not reveal how the decisions were made about relevance of studies and the validity of the included studies.

Changes in the manuscript: (52-53) This narrative review is based on a selection of studies relevant to our theme of interest. No specific criteria for inclusion and exclusion have been pre-defined (Collins & Fauser, 2004).

Collins, J.A. & Fauser, B.C.J.M. (2004) Editorial: balancing the strengths of systematic and narrative reviews. Human Reproduction Update, 11 (2), 103–110. https://doi.org/10.1093/HUMUPD/DMH058
